# “It Really Is an Elusive Illness”—Post-COVID-19 Illness Perceptions and Recovery Strategies: A Thematic Analysis

**DOI:** 10.3390/ijerph192013003

**Published:** 2022-10-11

**Authors:** Gerko Schaap, Marleen Wensink, Carine J. M. Doggen, Job van der Palen, Harald E. Vonkeman, Christina Bode

**Affiliations:** 1Department of Psychology, Health and Technology, University of Twente, 7522 NB Enschede, The Netherlands; 2Department of Health Technology and Services Research, Technical Medical Centre, University of Twente, 7522 NB Enschede, The Netherlands; 3Clinical Research Centre, Rijnstate Hospital, 6815 AD Arnhem, The Netherlands; 4Department of Epidemiology, Medisch Spectrum Twente, 7512 KZ Enschede, The Netherlands; 5Section Cognition, Data and Education, University of Twente, 7522 NB Enschede, The Netherlands; 6Department of Rheumatology and Clinical Immunology, Medisch Spectrum Twente, 7512 KZ Enschede, The Netherlands

**Keywords:** post-COVID-19 syndrome, long COVID, COVID-19, illness perceptions, self-regulation model, recovery, qualitative, reflexive thematic analysis

## Abstract

A substantial number of patients report persisting symptoms after a COVID-19 infection: so-called post-COVID-19 syndrome. There is limited research on patients’ perspectives on post-COVID-19 symptoms and ways to recover. This qualitative study explored the illness perceptions and recovery strategies of patients who had been hospitalised for COVID-19. Differences between recovered and non-recovered patients were investigated. Semi-structured in-depth interviews were held with 24 participating patients (8 recovered and 16 non-recovered) 7 to 12 months after hospital discharge. Data were interpreted using reflexive thematic analysis. Four overarching themes were identified: (I) symptoms after hospital discharge; (II) impact of COVID-19 on daily life and self-identity; (III) uncertainty about COVID-19; and (IV) dealing with COVID-19. Formerly hospitalised post-COVID-19 patients seem to have difficulties with making sense of their illness and gaining control over their recovery. The majority of non-recovered participants continue to suffer mostly from weakness or fatigue, dyspnoea and cognitive dysfunction. No notable differences in illness beliefs were observed between recovered and non-recovered participants

## 1. Introduction

A substantial number of patients with COVID-19 continue to experience chronic complaints after the acute phase, impacting their health, daily functioning, and health-related quality of life (HR-QoL) [1]. As of April 2022, an estimated 2.8 percent of all UK citizens experience lingering COVID-19 consequences four weeks after first being infected [2]. Furthermore, it is estimated that 10 to 35 percent of COVID-19 patients develop post-COVID-19 syndrome; for hospitalised patients this might be as high as 85 percent [3]. Post-COVID-19 syndrome—also referred to as long-COVID—is a condition occurring three months after a confirmed or suspected SARS-CoV-2 infection, lasting at least two months, with symptoms that cannot be explained by alternative diagnoses [4]. Commonly reported symptoms include fatigue, weakness, dyspnoea, cognitive dysfunction, and anosmia [5,6,7]. Previous studies have shown that, for example, fatigue has a significant impact on HR-QoL and the general health status of post-COVID-19 patients [8]. However, little is known about how patients experience COVID-19 sequelae in their daily life, or how they make sense of and cope with their illness and recovery process.

Patients’ perceptions, outcome expectations and coping with illness are generally influenced by how patients interpret their illness cognitively and emotionally. One useful framework to understand how COVID-19 patients experience their illness and recovery efforts might be the Common-Sense Model of Self-Regulation (CSM) [9]. According to this model (Figure 1), patients experiencing health complaints develop cognitive beliefs representing their illness. These can be considered in five dimensions: identity (name and presence of associated symptoms of condition), attributed cause, expected or experienced consequences, perceived control, and the timeline (acute, cyclical, or chronic)*,* of the illness. These cognitive representations, along with emotional responses to the illness such as anxiety, determine how people cope with their illness, i.e., by pursuing or avoiding treatment or recovery processes. The CSM has been found useful for investigating patient understanding and the course and impact of many different chronic conditions, such as Chronic Obstructive Pulmonary Disease [10], mental disorders [11], illness-related fatigue [12], and chronic illness in general [13].

There is limited research on how patients experience post-COVID-19 syndrome. Early studies of post-COVID-19 explored themes such as experiences of post-COVID-19 symptoms, the psychological impact, the healthcare system, and the difficulties accessing care experienced by non-hospitalised patients, who often did not have a confirmed SARS-CoV-2 infection [14,15,16]. However, none of these studies investigated post-COVID-19 illness representations and illness experiences of recovered patients in comparison to non-recovered post-COVID-19 patients.

Furthermore, research on how COVID-19 patients cope with their illness and recovery process is lacking. While many patients recover quickly from acute COVID-19, a substantial number do not. The reasons for this are still unknown. The discrepancies between illness perceptions and recovery strategies patients utilise might play a role. This study compares illness perceptions and self-reported recovery activities to explore the role of patient perceptions in the process of (non-)recovery after acute COVID-19. Information about these perceptions and coping strategies can guide healthcare support and future research.

The aim of this descriptive qualitative study was to explore the persistent symptoms, illness perceptions and recovery strategies of patients after hospital discharge after a SARS-CoV-2 infection. In addition, differences in experiences between recovered and non-recovered patients were investigated.

## 2. Materials and Methods

### 2.1. Design and Recruitment

Qualitative methodology via semi-structured in-depth interviewing was used to explore patients’ experiences and illness perceptions of post-COVID-19 syndrome. This design was chosen to better explore the richness and diversity of patient perspectives on and understandings of a novel disease. Ethical approval was granted by the Medisch Spectrum Twente Institutional Review Board (K20–30) and the Ethics Committee Behavioural, Management and Social Sciences of the University of Twente, the Netherlands (210799).

The present study was part of an ongoing project on the long-term impact of COVID-19 hospitalisation. At its centre is a cohort study investigating patients after being hospitalised due to confirmed SARS-CoV-2 infection. Participants in the present study were recruited via purposive selection based on the 3-month interval after hospital discharge from the cohort study. Forty-two patients were preselected based on self-reported health change compared to one year ago (prior to COVID-19 hospitalisation) as assessed with the Dutch SF-36 [17]. Patients were considered as recovered when they scored ≥50 (similar to or better than a year ago) or considered as non-recovered with a score <25 (much worse than a year ago). Ten recovered participants were randomly selected from a pool of 28. All 32 non-recovered patients were selected for an interview. During the interviews, self-reported recovery status was reassessed based on responses about the participant’s perceived current health.

Inclusion criteria were (a) having been discharged from hospital after PCR-confirmed acute COVID-19; (b) ≥18 years of age; and (c) proficient in Dutch. No exclusion criteria were used. Selected participants were invited by mail, along with a detailed patient information sheet and informed consent form.

### 2.2. Data Collection

Interviews were conducted either in-person at the participant’s home or via an online meeting, according to the patient’s preferences. The latter option was encouraged to limit virus exposure. All interviews were conducted one-on-one by two researchers (GS or MW) between June and October 2021. Participants were unfamiliar with the interviewers. Consent for participation and digital recording was (re)affirmed at the start of each interview.

A semi-structured interview guide (Figure 2) was developed with open questions, partly informed by the CSM constructs but broadened to stimulate participants to tell their own stories. Questions concerned any patient experiences since hospital discharge, including both ceased and ongoing complaints. A small pilot test with five non-hospitalised participants was run to optimise the prompts.

Data collection ceased after saturation had been reached. Audio recordings were transcribed verbatim by an AI-driven automatic transcription service and subsequently checked and edited by the interviewer. Participants’ age, gender, and self-reported height and weight were gathered from the cohort study. Relevant pre-existing health conditions were retrieved from the hospital medical records with the participant’s approval. As not all factors in developing (post-)COVID-19 are known, any health condition regardless of recency was reported.

### 2.3. Data Analysis

Transcripts were imported into ATLAS.ti version 9 for analysis. Data were interpreted via a reflexive thematic approach [18]. First, an initial codebook was created inductively by the interviewers (GS and MW), based on six transcripts (two recovered and four non-recovered participants). Secondly, all interviews were coded by each interviewer using this codebook, which was iteratively checked and updated throughout the data generation process. Next, the coded transcripts were reviewed by one researcher (GS) and inductively categorised into candidate themes and subthemes. The subthemes were then deductively compared to the CSM, after which the candidate themes and interpretation were discussed by three members of the research team (GS, MW and CB, health psychologists).

The interviewers (GS and MW) had limited experience in qualitative research. Analysis was strengthened by repeated close reading of transcripts and regular discussions between the two interviewers and a senior researcher (CB) who has extensive experience in qualitative analysis. The progress and final results were periodically discussed with the whole multidisciplinary research team.

## 3. Results

The final sample consisted of 24 participants (Table 1). Four of the 10 selected recovered patients and 22 of the 32 selected non-recovered patients agreed to participate. One recruited non-recovered patient was excluded because of insufficient Dutch proficiency and one non-recovered participant dropped out due to non-COVID-19-related health issues. Four participants previously selected as not-recovered expressed during the interviews to consider themselves to be recovered from COVID-19 sequelae and were subsequently categorised as recovered. Thus, the final sample consisted of 8 recovered and 16 non-recovered participants.

Participants’ ages ranged from 46 to 76 (mean 61.8 ± 8.5) years, 71% was male (*n* = 17) and 67% (*n* = 16) had one or more pre-existing health conditions. Three patients (all non-recovered) had been admitted to the intensive care unit. Patients had been discharged from the hospital between September 2020 and February 2021 and were interviewed between 7 and 12 months after hospital discharge. None had been vaccinated against COVID-19 prior to hospitalisation as vaccination was not yet available. However, most got vaccinated when it became available to them before the interviews.

The interviews lasted between 17 and 87 (median 64) minutes. Thematic analysis resulted in four themes: symptoms after hospital discharge, impact of COVID-19, uncertainty about COVID-19, and dealing with COVID-19. Illustrative data in the form of quotations are provided with patient recovery status: NR for non-recovered and R for recovered.

### 3.1. Symptoms after Hospital Discharge

Participants experienced a range of symptoms while recovering from COVID-19 up to the time of interviewing, categorised into three subthemes: somatic symptoms, cognitive symptoms, and psychological symptoms.

#### 3.1.1. Somatic Symptoms

All participants experienced weakness in the body, fatigue, or both at a certain point in time following the acute COVID-19 phase. While some participants noticed some gradual improvement since their hospital discharge, most considered loss of energy and fatigue as ongoing complaints. Out of the eight recovered participants, a majority still experienced a loss of energy (*n* = 5) and fatigue (*n* = 1) at the time of the interview. Conversely, 15 out of the 16 non-recovered participants still experienced these symptoms. One participant characterised these complaints as:
*“I’m easily fatigued, I’m easily out of energy. […] I used to do strength training and cardio [exercises] at the gym. But those strength exercises… I already experience difficulties when climbing stairs.”**(P22NR, M)*

Seven non-recovered participants experienced problems with sleep. Two of them mentioned that because of difficulty sleeping at night, they needed to take naps during the day.

Of the 17 participants who experienced dyspnoea after hospital discharge, 14 (13 non-recovered) still do. Most patients described their dyspnoea as being aggravated by exertion or humidity. In addition, several participants linked breathing problems to becoming fatigued.


*“It’s like a tightness in the upper airways, like a brick laying on your chest. You need to breathe that away. That’s that shortness of breath you feel.”*

*(P20NR, F)*


Additionally, six non-recovered participants described having adopted disordered breathing habits (‘breathing wrongly’) or having to relearn how to breathe ‘normally’ (i.e., take deep breaths) and automatically.

Eleven participants experienced pain or discomfort after hospital discharge, still persisting in nine non-recovered participants. Localised pains were headaches, chest pain and pain of the diaphragm, which was often attributed to ‘breathing wrongly’ or coughing. In addition, pain in the legs was mentioned by two non-recovered patients, while one patient experienced no feeling in the legs for several months until recovery.

A small subset of participants described various sensory complaints. Eight participants continued to experience loss of or changes in taste, smell, or appetite after the acute phase. Additionally, seven participants noticed a deterioration in their sight and hearing. Three participants developed tinnitus. Finally, four participants reported a tendency towards dizziness when getting up or during exertion. While more non-recovered participants reported sensory complaints, one recovered participant also continued to experience sensory issues.


*“Certain flavours and smells are wrong. [The taste of] peanut butter is off. […] Coffee beans don’t smell right.”*

*(P12R, M)*


In addition to post-COVID-19 symptoms, one reason why some participants did not consider themselves to be recovered is due to developing diseases after COVID-19, such as lung fibrosis, asthma, allergies, or a weakened immune system (*n* = 5) or worsening of pre-existing conditions (*n* = 3).

#### 3.1.2. Cognitive Symptoms

Eleven participants (10 non-recovered) experienced varying degrees of attention and concentration problems. Most reported an inability to concentrate or focus, while one reported feeling less alert and two not being able to multitask anymore. Only one participant reported having recovered from these complaints, which he characterised it as:
*“I just couldn’t manage anything. I couldn’t add anything together. My brain abandoned me for a long time.”**(P4R, M)*

In addition, 10 participants reported memory problems or confusion. Common complaints were problems with remembering names or words (*n* = 4), general forgetfulness (*n* = 4), and absent-mindedness or inability to think clearly (*n* = 5). All but one participant were non-recovered and still experience these complaints.


*“I noticed that I now have memory problems. […] I am 66, I’m allowed to forget things now and again. But I often just cannot remember the name of something. […] That has really worsened since COVID.”*

*(P8NR, F)*


#### 3.1.3. Psychological Symptoms

Eight participants described feeling down or less cheerful. Additionally, seven participants experienced problems with emotion regulation, namely being quickly ‘emotionally moved’ or irritated. Psychological problems were especially mentioned by non-recovered patients.


*“I‘ve got more emotional. […] I think I’ve cried for the last time 15 or 20 years ago. Now, I think I do so about once a week.”*

*(P14NR, M)*


Overall, participants described a variety and combination of physical, cognitive, and psychological complaints due to COVID-19 after hospital discharge. The vast majority of non-recovered participants continue to suffer mostly from weakness or fatigue, dyspnoea and cognitive dysfunction.

### 3.2. Impact of COVID-19

A common theme pertains to perceptions of the impact COVID-19 sequelae have. Subthemes are consequences and identity.

#### 3.2.1. Consequences

Participants, mostly non-recovered, described the impact COVID-19 sequelae had on daily life, ranging from not being able to do their job as usual or problems with starting the day (relatively light impediment) to inability to do anything at all (huge impediment). Impediments were mostly attributed to fatigue, weakness, and cognitive dysfunction. Furthermore, these were often linked with depressed moods and uncertainty by participants.

Light impediment:
*“I’ve been living more inwardly. I don’t talk as much as I used to do [due to dyspnoea].”**(P10NR, M)*

Huge impediment:
*“Being tired, moving, headaches, breathing: that shortness of breath. No strength in the legs. Yes, the limitations in daily life… [Nothing] is like it used to be. Everywhere, I meet my own limits.”**(P22NR, M)*

Additionally, participants discussed emotional reactions to their illness, lack of recovery or their bodies ‘letting them down’. Common reactions were anger, frustration, disappointment, and incomprehension.


*“I was frustrated, because due to the COVID, I couldn’t hike and run anymore. And if you do that every day for 15 years, that’s part of your life and you miss that every time.”*

*(P1R, M)*



*“Being mad at yourself, and disappointed, and having very high expectations of yourself. Not living up to it. Feeling guilty towards others, even though you don’t have to. Still, you aren’t yourself. At work, you think: you’ve abandoned them. Yes, those kinds of feelings. And… I’m young, I’m 48, and why for God’s sake?!”*

*(P5NR, F)*


#### 3.2.2. Identity

Post-COVID-19 affected how participants thought of themselves. While only some participants (all non-recovered) identified themselves as a post-COVID-19 patient, many acknowledged their identity was changed due to their illness.


*“[Illness] changes you. Yeah, you notice that it changes you.”*

*(P7NR, F)*



*“Sadly, I’ve moved to the group of ‘long-COVID patients’, so I have very recognisable complaints that I hear all around me, like ‘those belong to long-COVID’.”*

*(P16NR, M)*


Participants often compared their current health or physical state to a previous self-image, mostly expressing a desire to revert to ‘their own self’ again. However, recovery did not mean restoration to one’s pre-illness self-image. One recovered participant expressed not being ‘his old self’. Only one participant described feeling completely recovered.

Meanwhile, participants reported issues with a lack of understanding or recognition of post-COVID-19 syndrome by others. People did not understand how the patient could be so ill while most recover quickly from COVID-19. Moreover, participants noted that COVID-19 sequelae cannot be observed externally. For example:
*“From the outside I appear to be normal. I have no disabilities. So they think: ‘there he is, he’s healthy’. While you aren’t healthy, while you cannot function as normally.”**(P14NR, M)*

Additionally, participants became aware of their ageing. Several non-recovered participants described feeling older because of their illness. Additionally, both recovered and non-recovered participants mentioned that their age hindered recovery, especially the recovery of their physical fitness. Interestingly, two relatively young recovered participants (ages 49 and 58) attributed the lingering weakness to their age rather than to lack of (total) recovery. COVID-19-related ageing was expressed as:
*“It’s just like I’ve become an old man. Yes, I am old, but in the months of COVID, I’ve deteriorated by years.”**(P24NR, M; age: 62)*
*“Physically, I’ve aged. […] And my memory, that’s also annoying, because [my] children [ask] me: ‘do you have dementia?’ Someone else says: ‘that’s just your age.’ But yeah, I got it from COVID.”**(P8NR, F; age: 65)*

### 3.3. Uncertainty about COVID-19

This theme describes participants’ uncertainties concerning their illnesses. Subthemes are knowledge, cause, and timeline.

#### 3.3.1. Knowledge

Participants remarked how their future is uncertain, especially due to a lack of medical knowledge of this new illness. Moreover, there was anxiety about the (possible limited) extent of their recovery; whether they would even recover at all. Not only did participants themselves not know what to expect, but clinicians were also unable to provide satisfactory answers. Additionally, two recovered and one non-recovered participant remarked that it is not even clear whether they had ‘recovered from COVID-19’, as there are no tests available to know for certain.


*“The stupid thing is, with the coronavirus they say: ‘it’s just in the lungs’, but it’s not just the lungs. What does this [virus] do in your body? Nobody knows. What did it do? No idea. Are my muscles so affected that they’ve burned out? Have my joints been adjusted? Is it just the lungs? Is it your heart? What’s been roughed up? Even the specialists don’t know that. It could be years before we can say anything definitive about that.”*

*(P18NR, M)*


In trying to make sense of their illness and recovery, participants often compared it to previous illness experiences, most commonly influenza, but these comparisons failed. Furthermore, they wondered why *they* had to get ill or why they took so long to recover. This was especially true for participants without prior health issues.


*“That’s one of the things I’ve noticed about COVID: the emotion and also the physical response cannot be understood in advance. It really is an elusive illness.”*

*(P2R, M)*



*“It’s just like COVID isn’t just in the lungs, but in your entire body. Like it’s in your head as well. […] It’s also not like the flu. It’s an entirely different disease. If you’ve got it severe, you cannot compare it with anything. […] Everything is too much! And thinking, walking, talking, picking up something… It’s unimaginable!”*

*(P20NR, F)*


#### 3.3.2. Cause

Participants (mostly non-recovered) discussed ambiguities in the cause of their complaints. They wondered whether the persistent complaints were due to COVID-19, comorbidity, or both. Other explanations were medication, lifestyle, and age.


*“The fatigue, well, is that because of the fibrosis or because of COVID?”*

*(P19NR, F)*


Participants, recovered and non-recovered, described being fearful of reinfection with SARS-CoV-2, and being much more careful concerning prevention. Being vaccinated against COVID-19 relieved some of their anxiety.


*“Maybe you get COVID again. [… But] you have been vaccinated and have had COVID. If you develop it again, it should not be as severe.”*

*(P6R, M)*


#### 3.3.3. Timeline

Almost all participants agreed that recovery from COVID-19 was—against initial expectations—a very slow (“taking too long”) and often laboured process. While some participants mentioned seeing a bit of progress—often faster in the early stages—and expected to be near full recovery, a few others were pessimistic and considered they would never recover again. Moreover, especially non-recovered participants reported this lengthiness to be frustrating or difficult to accept. Participants described their recovery proceeding in small steps, often with relapses. This was attributed by some to fatigue or age.


*“[Recovery goes] very slow, very, very slow. A step forwards and a little bit backwards. And then some improvement and when you think you’re better, then suddenly a day or two, three, it doesn’t work.”*

*(P17NR, F)*



*“I don’t see me improve the [next] three months by much. After that, I hope to get better. I must take into account—I need to be honest—that my age won’t help. I will never be as fit as I used to be. I realise that, but it needs to be better than it is now.”*

*(P10NR, M; age: 74)*


### 3.4. Dealing with COVID-19

The final theme pertained to recovery strategies (support and individual strategies; see also Table A1) and attitudes and perceptions (experienced control over recovery and emotional coping).

#### 3.4.1. Support

Participants made use of several types of support from professional and informal sources. Physicians were visited for check-ups and monitoring, education, and advice (informational support). In addition, participants visited physiotherapists, occupational therapists, and rehabilitation centres.

Barriers to healthcare were also experienced. Some participants described not knowing what aid they needed or where to find it. This included not getting any information or referrals from the hospital or general practitioner (GP), nor any check-ups in the months after hospital discharge.


*“You didn’t get support. Not from your GP. Hospital was too busy with other things.”*

*(P4R, M)*


Furthermore, participants were dependent on healthcare insurance financing. Other limitations were due to personal circumstances or preferences, such as being restricted by the COVID-19 regulations or not wanting the help:


*“I kind of hate taking other people’s help. Even though they are professionals and get paid to do it. But, no, I’m not someone who’s dependent on others, at least, I never used to, and I prefer not to. I prefer managing everything on my own.”*

*(P10NR, M)*


Many participants received instrumental support for activities of daily life, such as household chores and groceries, mainly just after hospital discharge. This support was provided by professional homecare and domestic workers, as well as by spouses, family, and broader social networks. Informal aid was received with gratitude, although accepting care was often difficult as some participants were not used to being dependent on others.


*“In recent years I’ve [lived] always on my own. Then it’s nice to do things together, but I found it very difficult to ask acquaintances for help. And I learned to do that, to take that step, yes. Otherwise, you don’t get any food [laughs].”*

*(P24NR, M)*


Additionally, these private social networks also offered emotional support. Some participants visited (online) peer support groups and websites for informational support and recognition. For some, this quickly became ‘too much’ or merely induced unhappiness due to a focus on symptoms.

No relevant differences concerning support between recovered and non-recovered participants were found.

#### 3.4.2. Individual Strategies

Individually employed strategies entailed physical activity, pacing, and living healthily and positively. Participants described increasing their physical activity to rebuild their general fitness and strength through (unsupervised) exercise and by taking up activities of daily life. Almost all recovered participants explicitly stated that ‘taking up the old life’ was an important or the primary strategy.

Most participants stated that rebuilding their fitness went slowly and that taking up physical activity was often hindered. Reasons for this immediately after hospital discharge were still being contagious, being unable to exercise due to lack of breath or strength, or being advised against ‘overburdening the lungs’. The remaining barriers were persisting physical complaints or comorbidity.

Pacing was also considered to be important by participants. Most non-recovered participants expressed the need to spread out their exertion both over the day and over the week, limiting social activities, and being aware of both exertion and relaxation to prevent post-exertional malaise. Often, pacing was discussed as ‘returning to life as normal’, including going back to work or social participation. However, pacing was seen as a challenge, due to fatigue during activities and pushing their limits:
*“Don’t avoid physical activities, but be smart in what you do and don’t do. And just leave some things to others and don’t do it when you know it actually is too much. But yeah, every time, push up against the limits of your ability to see where that boundary lies and to stretch it a bit. I feel like I do as much as I can.”**(P16NR, M)*

Healthy living in general was mentioned as an individual strategy by non-recovered participants. They described eating healthily, using dietary supplements, drinking less alcohol, and “avoiding smoke”.

Positive attitudes regarding recovery were considered helpful by both recovered and non-recovered participants. Perseverance in recovery efforts was described as being pragmatic (‘doing nothing does not help’) and as being a character trait. Additionally, participants mentioned that staying optimistic and keeping faith in recovery was essential. Seeing improvements helped with keeping hope for total recovery.


*“I think it’s wise to just stay positive in these kinds of situations, even if it (…) is difficult when you’re in the hospital, because you have no control. But after that, every little step is a step, and every step forwards, however small, is something positive.”*

*(P1R, M)*


#### 3.4.3. Experienced Control over Recovery

Participants perceived their ability to control their recovery from COVID-19 as mixed. Interestingly, no clear differences between recovered and non-recovered participants were observed regarding experienced control. Some participants, mostly non-recovered, perceived recovery to be out of their hands, primarily due to seeing recovery simply as a matter of time: the body just has to restrengthen itself. While some reported that their recovery strategies were contributing to recovery, other participants did not always know what to do specifically.


*“I personally had no control over the recovery process, no, because slowly, you just return to your normal routine.”*

*(P6R, M)*


Perceived barriers to control over recovery were co-occurring medical conditions or relapses in illness. Some participants described overstretching themselves in their recovery efforts, sometimes leading to major setbacks.


*“I bit off more than I could chew. […] I wanted more. But the body, both mentally and physically, calls you back if you do too much. ‘Start again from the beginning.’ So I relapsed completely, and that was a shame.”*

*(P4R, M)*



*“And that’s of course the danger: wanting to recover too fast and instead of building up, you break down [your recovery progress]. That’s a pitfall.”*

*(P21NR, M)*


However, most participants perceived having at least some control, mostly due to taking up their recovery strategies. For some, this meant acting in accordance with advice from clinicians. In contrast, some participants conveyed to be themselves the main drive for recovery. Control was exerted through physical activity—especially doing exercises—and by individually conducting research on how to stimulate recovery.


*“For the main part, I had things under control. I had my own plan. But I needed and got help with that.”*

*(P11R, M)*


#### 3.4.4. Emotional Coping

Participants discussed how they coped with their emotional struggles with slow recovery. These were mostly, but not exclusively, addressed by non-recovered participants. They described using downward social (to other patients) and temporal (to a previous health state) comparison as coping strategies.


*“But I shouldn’t complain because a lot of people died from COVID-19. Here in the neighbourhood as well, acquaintances from my own town, a few of them passed away.”*

*(P10NR, M)*


A few non-recovered participants discussed acceptance, mentioning that they either needed to accept their situation or had already done so.


*“Letting go is the biggest art of life. That’s my experience. In the beginning I had the time: ’14 days and then we’ll see again’. I just let everything go, let it pass me, and I will see where it will end.”*

*(P19NR, F)*


Other coping strategies were positive framing and benefit finding. Considering their illness in positive terms helped some participants to cope. Rather than focusing on disability, some patients considered what they could still do. Others considered the positive consequences of their illness. Common reactions were reassessing the importance of one’s health and balance in life, improvement or recognition of social contacts, having learnt something from recovery, and the experience itself. Only non-recovered participants mentioned benefit findings.

## 4. Discussion

This study explored persistent symptoms, illness perceptions and recovery strategies of patients after SARS-CoV-2 infection-related hospitalisation. Participants reported being heavily affected in their daily life and identity, experiencing many uncertainties, and having a limited sense of control over their recovery. No notable differences were observed between recovered and non-recovered participants in these aspects. Participants experienced diverse persistent symptoms such as fatigue, dyspnoea, and cognitive dysfunction, consistent with previous studies, e.g., [7,14].

This study aimed to gain insights into how patients make sense of post-COVID-19 syndrome using the CSM as a conceptual framework. Participants experienced difficulties in understanding their illness: lack of clear recovery trajectories and outcome expectancies were identified. Additionally, past illness experiences did not help their sense-making. In general, negative illness perceptions can lead to lowered HR-QoL, although this can vary between diseases. For example, illness beliefs and HR-QoL were found to be lower in post-Q-fever patients compared to patients with chronic diseases such as diabetes [19]. HR-QoL was low in this current study, as was previously observed in post-COVID-19 patients [6]. This might suggest that patients with illnesses that are difficult to explain and predict need additional support and education in developing understanding. This could improve HR-QoL, especially in this novel disease. For example, participants in the current study described experiencing uncertainty or anxiety and not feeling in control over their recovery. A patient education intervention for functional somatic syndromes (e.g., chronic fatigue syndrome [CFS] and fibromyalgia; diseases with symptoms common to post-COVID-19 syndrome) reduced anxiety and negative illness beliefs, and improved pacing [20]. Thus, gaining an understanding of the disease and recovery process can guide future post-COVID-19 interventions, e.g., on how to improve post-COVID-19 symptoms and HR-QoL.

The impact on patients’ identities should also be considered. Participants reported a negative impact on their self-image due to disrupted daily activities and self-conceptions as ill or dependent persons, which did not necessarily improve after recovery. Additionally, they described not feeling recognised as patients or a lack of understanding by others, suggesting potential stigmatisation. In previous studies [14,16,21,22], post-COVID-19 patients similarly reported experiencing a threatened or spoiled identity due to a disrupted daily life or sense of self, as well as shame, stigma and misunderstanding by others, especially because of “physically invisible deficits” [22]. Distorted self-concepts can aggravate problem behaviours and symptoms such as fatigue, but might be improved through cognitive bias modification of biased self-concepts, such as in an ongoing project on fatigue in breast cancer patients [23]. Stigmatisation and invalidation were reported to negatively affect the HR-QoL and health outcomes in patients with CFS and fibromyalgia [24]: illnesses that are often considered by non-patients to be psychological (‘only in the mind’), due to a lack of physical explanations. Like CFS and fibromyalgia, and as participants pointed out, post-COVID-19 syndrome is not observable from the outside (a hidden disability). Consequently, patients often need to explain and defend themselves. The provision of psychoeducation would be beneficial, e.g., learning ways to discuss their symptoms with others, as well as learning to accept that invalidation might happen. For healthcare professionals, it is important to be aware of the roles of stigma and invalidation; to tackle them as a part of patient self-management and to be conscious of implicit expressions of disbelief [25]. Future studies should investigate the extent of stigma and invalidation, as well as the impact of negative self-concepts in post-COVID-19 patients.

Many participants reported experiencing limited control over their recovery. Perceived loss of control is common in diseases that are relatively new and of which symptoms are difficult to explain medically, for example, chronic Lyme disease [26], CFS [27] and fibromyalgia [28]. Attention should be paid to the control patients experience in their recovery and over their complaints. Additionally, treatment and self-management suggestions should be tailored to individual experiences [29] and preferences, such as the desire for psychological support and diverse forms of physical exercise (walking, cycling, strength training). Furthermore, improving patients’ understanding of post-COVID-19 syndrome might increase their sense of control. In an experiment with healthy participants, Bhogal, et al. [30] found that emphasising certainty and controllability, including referring to the illness as ‘ongoing COVID-19 recovery’ instead of ‘long-COVID’, increased experienced personal control and recovery control, as well as outcome expectations. They suggested that clinicians should provide information that promotes self-efficacy and that indicates where to find additional healthcare support (e.g., psychological support or occupational therapy, which many participants where not familiar with). Thus, it seems likely that increasing certainty about and emphasising control over the illness (treatment) will positively influence recovery control, outcome expectations and potentially health and HR-QoL outcomes.

Additionally, pacing as a self-management strategy should be discussed with patients. Many participants expressed experiencing control through physical activity, such as doing exercises. However, this was often obstructed through complaints, especially by energy loss, fatigue, and cognitive dysfunction. This result is in accordance with previous findings [15,29]. Patients experience difficulties with overburdening themselves or pushing their limits, resulting in overexertion. Post-exertional malaise was previously found as a significant challenge in post-COVID-19 sufferers [31,32]. Energy and activity pacing could alleviate this, but advice should be tailored towards patients’ expectations and clinical status [29]. Furthermore, monitoring and (technological) support could help with managing post-exertional malaise [33]. Taking up physical activity was not sufficient for patients to recover, suggesting a need for multimodal strategies. Finally, cultivating positive attitudes towards the illness might be a useful self-management strategy. Some participants reported being strengthened by staying optimistic. Optimism is found to be beneficial for reducing pain [34], and might also be of help with other symptoms and improving HR-QoL. Hence, positive psychological interventions might benefit post-COVID-19 patients. For healthcare professionals, this means that it is important to develop holistic perspectives on post-COVID-19 syndrome; to let go of ‘pure’ unidisciplinary views and employ multidisciplinary interventions. Thus, supporting patients in the physical and psychological aspects of their illness and motivating them to try out and adhere to multiple interventions. Moreover, a holistic understanding of post-COVID-19 syndrome may help healthcare providers in guiding post-COVID-19 patients towards understanding their illness.

### Strengths and Limitations

This study comprised a carefully selected sample of formerly hospitalised COVID-19 patients. Although the sample was representative based on post-COVID-19 symptomology, the relatively high number of male participants might limit generalisability, as gender differences have been found in the presentation and prevalence of post-COVID-19 syndrome [35,36,37]. While more women tend to develop post-COVID-19 syndrome, men are more likely to be hospitalised with severe acute COVID-19 [38,39], which was reflected in this sample.

A strength was that by including both recovered and non-recovered participants, ambiguities and differences in illness perceptions and recovery efforts could be investigated. However, a self-assessed definition of ‘recovered’ was used, with most ‘recovered’ participants still reporting weakness. So far, no objective demarcation of post-COVID-19 recovery exists. Anecdotally, some participants described themselves as being recovered because they were able to work full time again, even though they were more fatigued at the end of the day and workweek compared to the period before the SARS-CoV-2 infection. Future studies should look into when and why patients consider themselves to be recovered and how these self-descriptions match the clinical results.

Moreover, a possible selection bias could exist as patients who recovered without too many troubles might not have been interested in participation. Another limitation—due to time constraints—is that participants were not provided the opportunity to review the transcripts to correct errors, meaning that participants’ beliefs and perceptions could have been misrepresented. Finally, the reflective use of the well-established CSM for the interview guide and interpretation strengthened the rigour without limiting the flexibility of analysis. This allowed for presenting patient perspectives using bottom-up categories and for comparing the findings with previous studies in illness perceptions of other chronic diseases to guide understanding of post-COVID-19 syndrome.

## 5. Conclusions

This study found that post-COVID-19 patients appear to have difficulties with making sense of their illness and gaining control over their recovery. No notable differences in illness beliefs were observed between recovered and non-recovered participants. The majority of non-recovered participants continue to suffer mostly from weakness or fatigue, dyspnoea and cognitive dysfunction. Based on these and previous findings, healthcare providers would do well to discuss uncertainties of the course of the illness with patients, promote self-efficacy in self-management and guide pacing and physical exercise, and be aware of potential stigmatisation and invalidation. With regards to healthcare policy, institutions should offer multimodal, holistic care packages, as complaints are not just physical in nature, but also may require psychological and social support. Not all patients seem to be acquainted with all available support possibilities.

## Figures and Tables

**Figure 1 ijerph-19-13003-f001:**
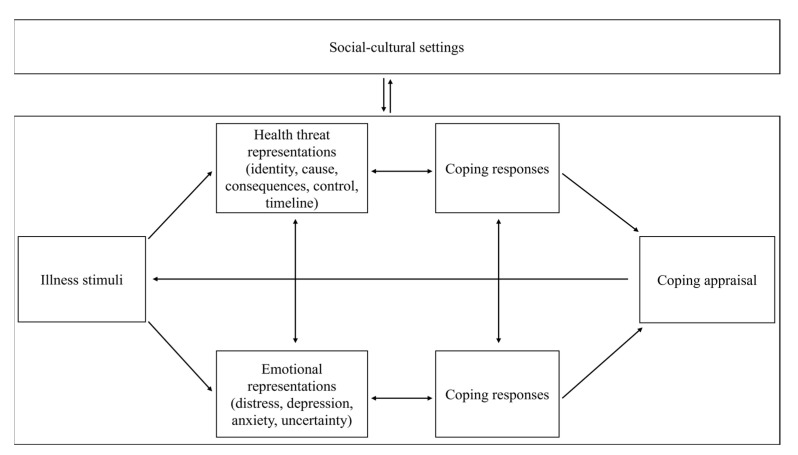
Common-Sense Model of Illness Representation.

**Figure 2 ijerph-19-13003-f002:**
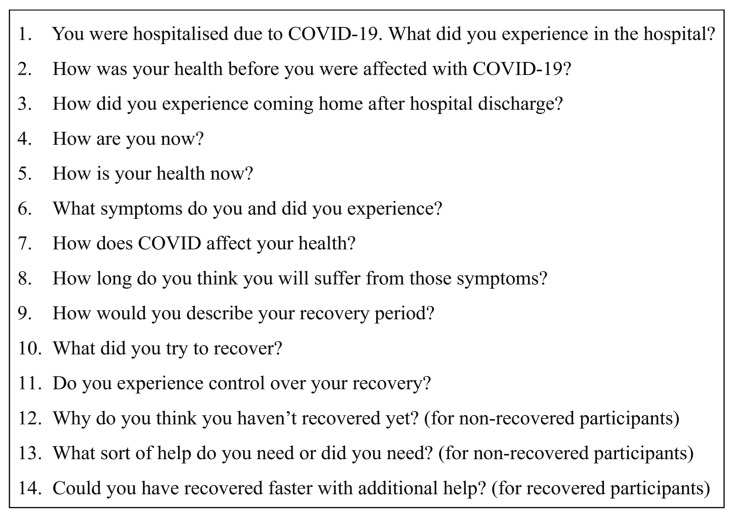
Interview guide.

**Table 1 ijerph-19-13003-t001:** Sample Characteristics.

	All (*N* = 24)	Non-Recovered (*N* = 16)	Recovered (*N* = 8)
Age, *M* (*SD*) years	61.8 (8.5)	60.8 (8.5)	63.8 (8.8)
Gender, *n* (%)			
Female	7 (29)	7 (44)	0 (0)
Male	17 (71)	9 (56)	8 (100)
Months since discharge ^a^, *M* (*SD*)	8.9 (1.4)	8.6 (1.2)	9.4 (1.7)
Body Mass Index (kg/m^2^), *M* (*SD*)	30.4 (5.6)	31.4 (6.2)	28.2 (27.7)
Number of pre-existing conditions, *n* (%)			
0	8 (33)	5 (31)	3 (38)
1	5 (21)	5 (31)	0 (0)
2	5 (21)	3 (19)	2 (25)
≥3	6 (25)	3 (19)	3 (38)
Pre-existing conditions ^b^, *n* (%)			
Anaemia	1 (4)	1 (6)	0 (0)
Cancer	2 (8)	1 (6)	1 (13)
Chronic respiratory disease	5 (21)	5 (31)	0 (0)
Cardiovascular disease	6 (25)	3 (19)	3 (38)
Diabetes mellitus	2 (8)	1 (6)	1 (13)
Dyslipidaemia	2 (8)	1 (6)	1 (13)
Gastrointestinal disease	3 (12)	2 (12)	1 (13)
Hypertension	8 (33)	4 (25)	4 (50)
Hypothyroidism	1 (4)	1 (6)	0 (0)
Musculoskeletal disorder	1 (4)	1 (6)	0 (0)
Obesity	11 (46)	8 (50)	3 (38)
Rheumatic disease	4 (16)	2 (12)	2 (25)
Urinary tract infection	1 (4)	0 (0)	1 (13)

Notes. Groups were based on self-assessed recovery from acute COVID-19 and COVID-19 sequelae; ^a^ Number of months between hospital discharge and interview; ^b^ Any pre-existing medical condition included in hospital medical records. *M* = Mean; *SD* = Standard Deviation.

## Data Availability

Study data are available upon reasonable request to the corresponding author.

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
