# Peer review of "“It Really Is an Elusive Illness”—Post-COVID-19 Illness Perceptions and Recovery Strategies: A Thematic Analysis"

_ijerph, 2022, doi:10.3390/ijerph192013003_

Round 1
Reviewer 1 Report
Thank you for the opportunity to review your paper, which discusses “It Really Is an Elusive Illness” -Post Covid-19 Illness Perceptions and Recovery Strategies.”
Overall, the paper is well written and provides a good overview of a well-designed study on a topic that is of concern for many people following a diagnosis of COVID-19. Understanding patients’ perceptions and experiences can be useful for health care providers in regard to how they explain COVID-19 recovery and advice on how to manage post COVID-19 symptoms. Findings are clearly presented and there is a clear interpretation of study results.
I will provide some comments under each heading of the paper as to where content could be strengthened, or where edits are required.
Abstract
The abstract clearly summarises the study . Line 19 – it would be good to clearly state that the study was qualitative in design – i.e. This qualitative study explored…..
Introduction
The introduction is well written and adequately sets the context for the study. Overall, there are contemporary references used. The aim is clearly presented.
Line 34 – the “s” needs to be removed from continues.
Materials and Methods
The materials and methods are clearly presented.
Line 103 –I would suggest that point d is removed as provision of the information sheet and consent is mentioned later.
Under data analysis – was there any opportunity to do member checking of the interview transcripts? If so, mention this and if not you could refer to this in the limitations, saying that undertaking member checking would further ensure that participants’ views were correctly represented.
Results
Overall, the results are interesting and well presented.
The sentence on lines 367-369 could be refined and the e.g. could be removed.
Discussion
A good discussion provided although some of the references to other studies that have focused on other diseases could be more clearly described. For example, line 445-447 – you could introduce this comparison more clearly and state that there is variation of illness beliefs and HR-Qol depending on the type of illness/disease.
On lines 449-452,you could link the findings of the study (reference 19) to the current study more clearly.
Line 458 – I suggest using a different word to addressed. Perhaps described would be better.
Lines 459-460 – you could expand a little more on the findings from references 14, 15, 20 and 21 and discuss how they are consistent with the current study.
Line 485 –use indicated not indicates. It would be useful to provide examples of where additional healthcare support could be found.
Line 496 – where you say advice should be tailored towards patients – it would be better if you clearly specified patient’s individual experiences and circumstances.
Line 513 – the word bias needs to be added after selection.
It would also be useful to include some discussion about how the study findings could be used by health care providers when working with post COVID-19 patients.
Conclusion
The conclusion summarises the main findings, but could be strengthened by having some final succinct comments on how the study findings are significant for practice, policy or further research.
Author Response
Dear reviewer,
Thank you for your kind words and clear remarks! They proved very helpful to us. We have explained our responses point-by-point (in red). Where we refer to lines, we have indicated the line numbers in the newly uploaded manuscript (with tracked changes).
- Abstract: The abstract clearly summarises the study . Line 19 – it would be good to clearly state that the study was qualitative in design – i.e. This qualitative study explored…..
We gladly have followed your suggestion. Moreover, we have indicated the study design (methodology used) in the title, as other reviewers have suggested.
- Introduction: The introduction is well written and adequately sets the context for the study. Overall, there are contemporary references used. The aim is clearly presented. Line 34 – the “s” needs to be removed from continues.
Thank you for pointing that out. We have removed the erroneous ‘s’.
- Materials and Methods: The materials and methods are clearly presented.
- Line 103 –I would suggest that point d is removed as provision of the information sheet and consent is mentioned later.
Good point, we have done so. In compliance with IJERPH guidelines, this information was already stated at the end of the article.
- Under data analysis – was there any opportunity to do member checking of the interview transcripts? If so, mention this and if not you could refer to this in the limitations, saying that undertaking member checking would further ensure that participants’ views were correctly represented.
Unfortunately, due to time restrictions, we did not check with the participants. We have identified this, per your suggestion, as a limitation in the strengths and limitations section (lines 580 – 583).
- Results: Overall, the results are interesting and well presented. The sentence on lines 367-369 could be refined and the e.g. could be removed.
We have revised this sentence (now lines 398-399); hopefully this is clearer.
- Discussion: A good discussion provided although some of the references to other studies that have focused on other diseases could be more clearly described. For example, line 445-447 – you could introduce this comparison more clearly and state that there is variation of illness beliefs and HR-Qol depending on the type of illness/disease. On lines 449-452,you could link the findings of the study (reference 19) to the current study more clearly.
We have endeavoured to comply with this. By adding to and reshuffling some sentences (lines 483-486), we have tried to argue why some diseases might be comparable to post-COVID in regards to illness perceptions. By first explaining why we think post-COVID-19 syndrome can be compared to diseases such as post-Lyme disease, chronic fatigue syndrome and fibromyalgia, later arguments become clearer. We also tried to make the connections to our findings more explicit, such as in lines 492-493.
- Line 458 – I suggest using a different word to addressed. Perhaps described would be better.
We agree and have incorporated this.
- Lines 459-460 – you could expand a little more on the findings from references 14, 15, 20 and 21 and discuss how they are consistent with the current study.
In lines 503-506, we have now described what these previous studies found about identity and stigma. As these are similar to what we have found, we hope the references are now clear.
- Line 485 –use indicated not indicates. It would be useful to provide examples of where additional healthcare support could be found.
In this sentence, ‘indicates’ should have referred to the information, not to the clinicians (or the authors of the cited study). We now have clarified this (‘information that indicates…’) and added some examples of support (line 534). Later on in the discussion, we have added a few sentences arguing that healthcare providers themselves might not be familiar with additional support, and we rather encourage them taking multidisciplinary, holistic perspectives on post-COVID-19 syndrome (lines 555 – 561). With this, we hope our point comes across more clearly.
- Line 496 – where you say advice should be tailored towards patients – it would be better if you clearly specified patient’s individual experiences and circumstances.
We have added ‘expectations and clinical status’ to this, and recommend readers to see Humphrey et al. (2021)’s discussion (line 547).
- Line 513 – the word bias needs to be added after selection.
‘Bias’ has been added so that it now reads ‘selection bias’.
- It would also be useful to include some discussion about how the study findings could be used by health care providers when working with post COVID-19 patients.
- Conclusion: The conclusion summarises the main findings, but could be strengthened by having some final succinct comments on how the study findings are significant for practice, policy or further research.
We thank the reviewer for these suggestions and the opportunity to make these ideas more explicit. As aforementioned, we have added a few sentences arguing for healthcare professionals to take holistic views about post-COVID-19 syndrome (and related interventions and treatment; lines 555 – 561) and also added a summary of our suggestions to the conclusion (lines 593 – 599).
Reviewer 2 Report
Overall, this was a nicely written and well-thought out evaluation of the qualitative experience of post-COVID-19 illness. I have relatively minor comments only.
* I found it difficult to see how the Common Sense Model had been applied specifically and it wasn't clear to me how the interpretation was different from other qualitative analysis. If the authors feel strongly that the CSM is a powerful interpretative tool, they need to make their arguments for this more obvious.
* The cohort for this study has an unusually high proportion of male participants. The authors do comment on this but I wonder if they have any insight as to *why* their population is so different?
* Although the authors make several clear statements that there were limited differences between the 'recovered' and 'not recovered' groups, the majority of the exemplar quotes were from the 'not recovered' groups. This made it difficult to confirm the assertions made by the authors. If the quotes offered were more evenly balanced between the two groups the results would be clearer to the reader.
* It would be helpful to identify the participants in more detail than at present. As the manuscript stands, the quotes presented could be from just two people. Some indication of multiple participant responses would be better, for example NR, M, 3; or R, F, 8.
* In the discussion it is mentioned that a number of participants who considered themselves 'recovered' were still displaying symptoms of weakness. Some discussion of what 'recovered' therefore means would be useful - either here or in the main results if the authors have the data to support this.
Author Response
Dear reviewer,
Thank you for your kind words and clear remarks! They proved very helpful to us. We have addressed your remarks point-by-point (in red). Where we refer to lines, we have indicated the line numbers in the newly uploaded manuscript (with tracked changes).
- I found it difficult to see how the Common Sense Model had been applied specifically and it wasn't clear to me how the interpretation was different from other qualitative analysis. If the authors feel strongly that the CSM is a powerful interpretative tool, they need to make their arguments for this more obvious.
For the purpose of our current undertaking, we ourselves would not use the word ‘powerful’. However, we think it has proved very useful by means of being a guideline when developing the interview guide (see line 115) as well as a guideline to report the bottom-up developed categories from the patient perspectives (our original codes) in a way that makes them easier to read and comparable to previous studies investigating illness perceptions of other chronic conditions. By doing so, we developed a better understanding of how these illnesses compare, and how they should be monitored and intervened in. We have now tried to argue this succinctly in the article as a strength (lines 585 – 587).
- The cohort for this study has an unusually high proportion of male participants. The authors do comment on this but I wonder if they have any insight as to *why* their population is so different?
Thank you for pointing out this excellent point. Most likely, this is due to our recruitment strategy, which involved recruiting participants (formerly hospitalised patients) from the overarching project to our current study. While more women than men develop post-COVID-19 syndrome, in general, men are more likely to develop more severe acute COVID-19 and hence are more likely to be hospitalised. Consequently, our ‘starting pool’ of candidates consisted of more men than women. We have added this explanation to the strengths and limitations section (lines 567 – 569).
- Although the authors make several clear statements that there were limited differences between the 'recovered' and 'not recovered' groups, the majority of the exemplar quotes were from the 'not recovered' groups. This made it difficult to confirm the assertions made by the authors. If the quotes offered were more evenly balanced between the two groups the results would be clearer to the reader.
We agree that the balance between quotations from the two groups is off and have added quotes from the recovered groups (or from the non-covered group to show the similarities or differences) to better represent the illness beliefs. However, in total, it currently still is unbalanced. This is because of a couple reasons. First of all, we have tried not to prolong the results section too much, so quotations were used sparingly (especially as Reviewer 4 suggested to shorten it). Secondly, the two groups are also not balanced equally (8 recovered vs 16 non-recovered participants), so the pool of potential quotes also is smaller. Thirdly, the remarks from recovered participants are not always that easy to present as participants focused on different aspects of (post-)COVID-19 in their narratives. Take for example impediments: non-recovered participants who still perceive to have a poor quality of life express more clearly what it is like to live with post-COVID-19, while recovered participants often only stated that this was poor(er) but told us more (and more detailed) about their recovery processes. We have opted to share the most illustrative quotations for these reasons, and hope the reviewer and editor agree with this.
- It would be helpful to identify the participants in more detail than at present. As the manuscript stands, the quotes presented could be from just two people. Some indication of multiple participant responses would be better, for example NR, M, 3; or R, F, 8.
Thank you for this suggestion, we have adjusted this. Now, we state the participant ID, recovery status and gender. We have purposefully left out age unless contextually necessary to keep the participants as anonymous as possible. If the reviewer or editor disagrees with this, we can further adjust this of course.
- In the discussion it is mentioned that a number of participants who considered themselves 'recovered' were still displaying symptoms of weakness. Some discussion of what 'recovered' therefore means would be useful - either here or in the main results if the authors have the data to support this.
That is an excellent question, but one we don’t have enough information about to answer. We have addressed this in the strengths and limitations section (lines 574 - 578). Based on anecdotal evidence (only a few participants mentioned this, as this was not one of our lines of questioning; at least the corresponding author also has heard this more than once outside of this project), we think that some patients think they are ‘well enough again’ to participate in society so that they ‘should consider themselves’ to be recovered. However, other patients, and healthcare workers, might disagree with this. More research on this topic is necessary.
Reviewer 3 Report
Good quality semi-qualitative review. Useful findings, described in a straightforward manner.
In terms of presentation, both subtitles and patients' quotes are in Italics. Perhaps it would be clearer to use underlining for subtitles.
Three small spelling or language errors
Line 258. What does 'I've moved to the host of Long Covid patients' mean?
Line 382. Should be 'healthily' , not 'healthy'.
513 a 'possible selection bias?'
I read in one place, the word 'gotten' which would not be UK English.
Author Response
Dear reviewer,
Thank you for your kind words and clear suggestions! They proved very helpful to us. We have addressed your remarks point-by-point (in red).
- In terms of presentation, both subtitles and patients' quotes are in Italics. Perhaps it would be clearer to use underlining for subtitles.
Thank you for pointing out the unclarity. In compliance with the IJERPH format, we have now reported the subtitles as (level 3) subheadings. This way, the distinction should now be clear. However, if the editor agrees with the reviewer that subtitles can and should be underlined, we would gladly do so.
- Three small spelling or language errors
- Line 258. What does 'I've moved to the host of Long Covid patients' mean?
- Line 382. Should be 'healthily' , not 'healthy'.
- 513 a 'possible selection bias?'
- I read in one place, the word 'gotten' which would not be UK English.
Thank you for showing us these errors. We have adjusted them all accordingly. With regards to host, we originally tried to translate the Dutch word leger as directly as possible. This would be ‘host’, both meaning a large group or gathering (but also have other meanings: of course 'one who accommodates or entertains' for host, while leger also can mean army in Dutch, in case you are interested in languages). However, this usage of host might be obscure and confusing, so now we have opted for the word ‘group’. We have looked up the context in which the participant expressed this idea; the main takeaway is that the participant sees himself as part of a group of patients with post-COVID-19 complaints.
Reviewer 4 Report
Dear authors, the manuscript is quite fluent, the methodology has no serious issues apart from the small number of participants, beyond the qualitative nature of the paper, but I can suggest these concerns to address
line 2 I suggest not to repeat illness. use syndrome for example. yes add "a thematic analysis"
Line 20 in light of the objective difference between hospitalized patients, I would clarify the previous concept of hospitalized patients
L23. I suggest first defining that it is a thematic analysis study
L24 Quality of life
Participants are few, however the speculative nature of the study suggests softening the conclusions with: it seems, it appears ..
44 Moreover: "Fatigue could result from many factors, including post-viral fatigue syndromes, respiratory muscle weakness, and possible general deconditioning linked to post-intensive care syndrome. In this context, fatigue has a significant impact on health-related quality of life and general health status of post-COVID -19 subjects, with a decrease of up to 69% of patients from 14 days to 3 months after infection and a reduction in the performance of activities of daily living ( ADL). ref: https://doi.org/10.3390/app12178593
78 qualitative study
90 any reference? protocol code? other paper or prepress?
94 is a result reported in the next section. put the exclusion criteria for the selection, instead of the selection itself (age? selected comorbidities? BMI?)
142 I would suggest shortening the results section.
518 awareness raising?
Author Response
Dear reviewer,
Thank you for your kind words and helpful comments! We have addressed your remarks point-by-point (in red). Where we refer to lines, we have indicated the line numbers in the newly uploaded manuscript (with tracked changes).
- line 2 I suggest not to repeat illness. use syndrome for example. yes add "a thematic analysis"
While we agree that the repetition of the phrase ‘illness’ is not ideal, we kindly like to point out that here it refers to two separate constructs, the first (from the quotation) being about the experience of the illness, the second about illness perceptions. Syndrome or disease would refer to different concepts.
Indicating the analytical design in the title is an excellent suggestion, which we happily follow.
- Line 20 in light of the objective difference between hospitalized patients, I would clarify the previous concept of hospitalized patients
For this point, we could not fully understand what the reviewer would like to see. The reference to hospitalised patients is just to indicate how we approached our (pre)selection and the overarching target group that we have sampled. For analytical purposes, we have focused on the differences between recovered and non-recovered, formerly hospitalised patients. We have explained this in more detail later in the Materials and Methods section. Due to word count limitations, we do not think it’s appropriate to expand on this in the abstract.
- I suggest first defining that it is a thematic analysis study
As per the suggestions of reviewer 1, we now have indicated the study design. As we agree that indicating the analytical approach also makes sense, so the reader knows what kind of results they may expect, we have made sure this was mentioned in the ‘methods part’ of the abstract and have tried to clarify this (line 23). Between that and the change in title, we hope this is now clear enough.
- L24 Quality of life
Here we report the theme, which talks about an aspect of quality of life, but is not similar. Hence, we have left ‘impact of COVID-19 on daily life and self-identity’ as is. If we have misunderstood the reviewer, we are open for clarification of course.
- Participants are few, however the speculative nature of the study suggests softening the conclusions with: it seems, it appears ..
Thank you for pointing this out to us. We have now tried to make clear that when we are talking about patients as a population, we are speculating by using the suggested words (e.g. line 26).
- 44 Moreover: "Fatigue could result from many factors, including post-viral fatigue syndromes, respiratory muscle weakness, and possible general deconditioning linked to post-intensive care syndrome. In this context, fatigue has a significant impact on health-related quality of life and general health status of post-COVID -19 subjects, with a decrease of up to 69% of patients from 14 days to 3 months after infection and a reduction in the performance of activities of daily living ( ADL). ref: https://doi.org/10.3390/app12178593
Thank you for that interesting article. We have included it in our introduction (lines 45 – 46).
- 78 qualitative study
We have now clarified the study design in the introduction (line 80). Thank you for bringing this to our attention!
- 90 any reference? protocol code? other paper or prepress?
Unfortunately we do not have those as of yet. So far, no (prepress) articles have been published using this project. We are working on registering the project, but this may still take a while.
- 94 is a result reported in the next section. put the exclusion criteria for the selection, instead of the selection itself (age? selected comorbidities? BMI?)
Here too, we are not entirely sure what the reviewer expects from us. Hopefully the following helps to clarify any questions the reviewer has. First, we have reported two different things. In the Methods section, we explain how we have recruited our participants by first making a preselection of ‘recovered’ and ‘non-recovered’ (former) patients. We have now indicated that this is a preselection (line 96). The preselection was mostly to make sure that we had enough participants to start the project. In our Results we describe how we have reassessed and finally categorized the two groups, using multiple questions and general answers from the participants during the interview (e.g. ‘how is your health now?’).
During selection, we only had inclusion criteria, as at the time, we did not know enough about predictors of post-COVID-19. We have now indicated in the article that we had no exclusion criteria (line 106). All characteristics served to describe the sample.
- 142 I would suggest shortening the results section.
Due to reviewers differing in their suggestions, with reviewer 2 seemingly wanting to expand on the Results section, and without indications on where or how to shorten the Results, we currently cannot comply with this suggestion.
- 518 awareness raising?
We have now expanded on the Conclusion and Discussion, so that it is more explicit what we would like for healthcare providers and policy makers to take away. Awareness raising has been included – albeit not so explicit – in our suggestions, i.e. being aware of potential stigmatization and invalidation; calling for multidisciplinary and holistic perspectives. If the reviewer has a different idea in mind, we would love to hear about it.
Round 2
Reviewer 4 Report
Dear authors, the methodological character has increased, it remains a small sample but with a qualitative facet